# Target Detection in Hyperspectral Remote Sensing Image: Current Status and Challenges

**Bowen Chen** [1,2,3], **Liqin Liu** [1,2,3], **Zhengxia Zou** [4] **and Zhenwei Shi** [1,2,3,*]

1   Image Processing Center, School of Astronautics, Beihang University, Beijing 100191, China; chenbowen@buaa.edu.cn (B.C.); liuliqin@buaa.edu.cn (L.L.)
2   Beijing Key Laboratory of Digital Media, Beihang University, Beijing 100191, China
3   State Key Laboratory of Virtual Reality Technology and Systems, Beihang University, Beijing 100191, China
4   Department of Guidance, Navigation and Control, School of Astronautics, Beihang University, Beijing 100191, China; zhengxiazou@buaa.edu.cn
*   Correspondence: shizhenwei@buaa.edu.cn

**Abstract:** Abundant spectral information endows unique advantages of hyperspectral remote sensing images in target location and recognition. Target detection techniques locate materials or objects of interest from hyperspectral images with given prior target spectra, and have been widely used in military, mineral exploration, ecological protection, etc. However, hyperspectral target detection is a challenging task due to high-dimension data, spectral changes, spectral mixing, and so on. To this end, many methods based on optimization and machine learning have been proposed in the past decades. In this paper, we review the representatives of hyperspectral image target detection methods and group them into seven categories: hypothesis testing-based methods, spectral angle-based methods, signal decomposition-based methods, constrained energy minimization (CEM)-based methods, kernel-based methods, sparse representation-based methods, and deep learning-based methods. We then comprehensively summarize their basic principles, classical algorithms, advantages, limitations, and connections. Meanwhile, we give critical comparisons of the methods on the summarized datasets and evaluation metrics. Furthermore, the future challenges and directions in the area are analyzed.

**Keywords:** hyperspectral remote sensing image; target detection; remote sensing

## 1. Introduction

Remote sensing techniques play a vital role in both military and civilian applications [1,2]. Hyperspectral remote sensing obtains the rich spectral information of ground objects, which provides unique advantages for probing and distinguishing various targets [3,4], and thus plays a crucial role in the remote sensing image processing field. As a result, hyperspectral target detection (HTD) has become a research hotspot in the field of hyperspectral image processing.

Hyperspectral target detection utilizes the spectral information of each pixel in the hyperspectral image (HSI) to determine whether the pixel belongs to a certain material. It can be divided into two categories: one is known as target detection and the other is anomaly detection. Target detection is the task of finding and localizing targets in a hyperspectral image given the reference spectrum of the target. The reference spectra are usually obtained from the spectral library or target pixels already identified in the scene. Typically, only one or a few reference spectra of the target are available. Anomaly detection marks the anomalous objects in HSI without the requirement of prior knowledge from the target spectrum. Since anomaly detection highlights the target without clarity of the interesting prior, it is not suitable for the targeted detection of objects of prior interest. In this paper, we focus on target detection with a reference spectrum given to mark certain targets and refer it to target detection later for brevity.

Benefiting from the accurate detection of certain targets, target detection is widely used in various fields. First, it is used to detect important military targets such as aircraft, ships, airports, oil tanks, and landmines, and is thus of great importance for military reconnaissance and strikes [5–7]. Second, in the field of forest science, it can be used for the detection of new leaf growth [8] and the monitoring of forest diversity and structure [9]. Third, in the field of mineral prospecting, hyperspectral target detection can be used for the detection of iron oxides [10] and the detection of minerals in geothermal prospect areas [11]. Finally, there are also a large number of applications in other civil fields, such as post-disaster rescue [12], gas detection [3], and precision agriculture [13].

Nowadays, target detection methods have been developed from various advanced techniques, including signal processing techniques, optimization techniques, and machine learning techniques. In recent years, the booming development of deep learning has injected new energy into the field. Although target detection methods have been extensively developed and explored in many application areas, challenges still exist in this field due to spectral variability, difficulty in the acquisition of the ground truth, etc. Therefore, a comprehensive overview of the current status and future challenges of hyperspectral target detection is crucial. We reviewed the previous review papers and found that most of them suffer from the following problems:

(1) Incomplete introduction. Earlier reviews have combed through traditional target detection methods [14–18], but they have not focused on the deep learning-based methods that have emerged in recent years and are becoming mainstream.

(2) Insufficient relevance. Although some recent related reviews contain some relatively advanced methods [19–21], they do not focus directly on the field of target detection, but broadly on hyperspectral image processing, which is not relevant enough. In addition, most of these reviews only list the advanced methods, and the summary and comparison of these methods are not satisfactory.

(3) Neglect of connections between methods. Most of the existing reviews only focus on the differences between the various methods and introduce each type of method independently, neglecting to explore the connections between different types of methods.

To address the incomplete introduction, we add the summary of sparse representation-based methods and deep learning-based methods to the traditional methods. To address the insufficient relevance, we focus exclusively on target detection and analyze the problems and challenges unique to the target detection task. To address the neglect of connections between methods, we analyze the differences and connections between various types of methods, and link the hyperspectral target detection methods together systematically.

In this paper, we focus on target detection and give a comprehensive review of it. We systematically summarize and categorize the existing methods and give a brief introduction to the representative algorithms. Meanwhile, we provide outlines of datasets and evaluation metrics for target detection as well as an outlook on future challenges in this paper. We hope the research in this paper will be useful to new researchers interested in the field of hyperspectral target detection and to those who want to further their research in hyperspectral target detection.

The remaining part of this paper is organized as follows: Section 2 provides a review of target detection methods. Section 3 introduces the details about datasets and evaluation metrics. Section 4 provides a comparison of the methods from the point of view of core ideas and experimental results. Meanwhile, we point out the future challenges and directions in Section 4. Finally, the conclusion is drawn in Section 5.

## 2. Target Detection Methods

In this section, we give a comprehensive review of the detection methods and divide them into the seven following categories: hypothesis testing-based methods, spectral angle-based methods, signal decomposition-based methods, constrained energy minimization (CEM)-based methods, kernel-based methods, sparse representation-based methods, and deep learning based methods. We first provide a general description of the basic ideas,

advantages, and disadvantages of these seven categories in Section 2.1. After that, in Sections 2.2–2.8, we provide specific descriptions of the seven categories of algorithms.

### 2.1. Overview

We first introduce one of the most classical hyperspectral target detection methods, the hypothesis test-based methods, in Section 2.2. Such methods model the hyperspectral target detection problem as a hypothesis testing problem and use the likelihood ratio as the basis for determining whether a pixel is a target or not. In this category, different modeling approaches can derive different forms of detectors. In Section 2.2, we also introduce the concept of data whitening and use this concept to relate the different detectors in the following subsections.

In Section 2.3, we introduced the spectral angle-based methods. The basic idea of this type of detector is to match each pixel in the HSI using a known reference target spectrum. This type of detector is easy to compute but has limited performance.

In Section 2.4, we model the hyperspectral target detection problem as a signal decomposition problem from a signal processing perspective. By modeling the different decompositions of the signal, we obtain different detectors. This type of detector is physically interpretable but often requires numerous prior information.

In Section 2.5, we construct the detector from the viewpoint of filtering and use an optimization-based approach to optimize the detector. Specifically, constraining the response of the target detector to the target spectrum minimizes the output energy of the detector. This class of detectors is called constrained energy minimization (CEM) based detectors. This type of detector is still widely used today because of its good suppression of the background.

However, the basic forms of the above detectors are linear. As hyperspectral data often have numerous nonlinear properties, it is therefore important to increase the nonlinear detection capability of the detectors. In Section 2.6, we introduce the kernel method to map the data to a high-dimensional space to remove the nonlinear properties in HSI. This approach can handle nonlinear data better but has a huge computational overhead.

In recent years, data-driven methods have gradually become a research hotspot for hyperspectral target detection due to the advantages of good detection performance and robustness. In Section 2.7, we introduce the sparse representation-based methods. Such methods reconstruct the target and the HSI by constructing a suitable dictionary and performing detection on the reconstructed results. This type of method is effective against spectral variability but relies on the construction of dictionaries. In Section 2.8, we introduce deep learning-based methods. These methods either detect the target directly by building a neural network model or first reconstruct the target and the HSI using the neural network model, and then detect it later using traditional target detection methods. Deep learning-based methods have high accuracy but suffer from problems such as a lack of data and limited interpretability.

Suppose a hyperspectral image can be arranged as a matrix $\mathbf{X} = [\mathbf{x}_1, \mathbf{x}_2, \ldots, \mathbf{x}_N] \in \mathbb{R}^{L \times N}$, where each column of $\mathbf{X}$ is an $L$ dimensional spectral vector $\mathbf{x}_i$, $N$ is the number of pixels, and $L$ is the number of wavebands. Suppose $\mathbf{d}$ is an $L$ dimensional column vector representing the target reference spectrum.

### 2.2. Hypothesis Testing-Based Methods

Target detection can be seen as a hypothesis-testing problem. Let $H_0$ represent that the target is absent, and let $H_1$ represent that the target is present. A common approach to solving hypothesis testing problems is to construct likelihood ratio testing. Let $f_0(\mathbf{x}_i \mid H_0)$ be the conditional probability density function of the observed spectrum $\mathbf{x}_i$ under the hypothesis $H_0$ and $f_1(\mathbf{x}_i \mid H_1)$ be the conditional probability density function of the observed pixel $\mathbf{x}_i$ under the $H_1$ hypothesis. Then the likelihood ratio can be defined as:

$$\Lambda(\mathbf{x}_i) = \frac{f_0(\mathbf{x}_i \mid H_0)}{f_1(\mathbf{x}_i \mid H_1)}. \tag{1}$$

Let the threshold be $\tau$. If $\Lambda(x_i) > \tau$, the hypothesis $H_1$, which indicates the presence of the target, is accepted, and if $\Lambda(x_i) < \tau$, the hypothesis $H_0$, which indicates the absence of the target, is accepted. In order to determine the likelihood ratio, two conditional probability density functions $f_0(\mathbf{x}_i \mid H_0)$ and $f_1(\mathbf{x}_i \mid H_1)$ need to be known. The forms of the conditional probability density functions are different under different forms of hypotheses.

Assuming that both hypotheses follow the Gaussian distribution and that their covariance matrices are equal, the hypothesis testing model can be denoted as:

$$
\begin{aligned}
H_0 \;&:\; \mathbf{x}_i \;\sim\; N(\boldsymbol{\mu}_b, \boldsymbol{\Sigma}) \\
H_1 \;&:\; \mathbf{x}_i \;\sim\; N(\boldsymbol{\mu}_t, \boldsymbol{\Sigma}),
\end{aligned}
\tag{2}
$$

where $\boldsymbol{\mu}_b$ and $\boldsymbol{\mu}_t$ represent the mean vectors of the background and target, respectively, and $\boldsymbol{\Sigma}$ represents the covariance matrix. Therefore, the likelihood ratio can be expressed as:

$$
\Lambda(\mathbf{x}_i) \;=\; \frac{\exp\left[-\frac{1}{2}(\mathbf{x}_i - \boldsymbol{\mu}_t)^\top \boldsymbol{\Sigma}^{-1}(\mathbf{x}_i - \boldsymbol{\mu}_t)\right]}{\exp\left[-\frac{1}{2}(\mathbf{x}_i - \boldsymbol{\mu}_b)^\top \boldsymbol{\Sigma}^{-1}(\mathbf{x}_i - \boldsymbol{\mu}_b)\right]}.
\tag{3}
$$

Taking the logarithm of $\Lambda(\mathbf{x}_i)$ and ignoring the constant term, we yield the following Matched Filter (MF) detector:

$$
\delta^{MF}(\mathbf{x}_i) \;=\; k(\boldsymbol{\mu}_t - \boldsymbol{\mu}_b)^\top \boldsymbol{\Sigma}^{-1}\mathbf{x}_i \;=\; \mathbf{w}_{MF}^\top \mathbf{x}_i,
\tag{4}
$$

where $k$ is a normalization constant. Since the detector shown in Equation (4) has the same form as the matched filter, it is called the MF detector.

The MF detector in this form requires access to prior background information, which is usually not readily available. To solve this problem, we can also use the hypothesis testing model in the following form:

$$
\begin{aligned}
H_0 \;&:\; \mathbf{x}_i \;=\; \mathbf{v} \;\sim\; N(0, \boldsymbol{\Sigma}) \\
H_1 \;&:\; \mathbf{x}_i \;=\; a\mathbf{s} + \mathbf{v} \;\sim\; N(a\mathbf{s}, \boldsymbol{\Sigma}).
\end{aligned}
\tag{5}
$$

where $\mathbf{s}$ in the above equation represents the pure target (or called endmember), indicating that the spectrum contains only one material. When the reference target spectrum $\mathbf{d}$ is also considered to be a pure target, the two are equivalent. In Equation (5), $\mathbf{v}$ represents noise and $a$ represents the abundance factor (the proportion of the endmember in the pixel). Since a is a ratio, $a > 0$.

While assuming that the target and background covariance matrices are the same, the above model additionally gives the two assumptions that the target is superimposed from the pure target and background and that the background has the same mean value under both $H_0$ and $H_1$ hypotheses. The MF detector derived from the above model can be denoted as [22]:

$$
\delta^{MF}(\mathbf{x}_i) \;=\; a\mathbf{s}^\top \boldsymbol{\Sigma}^{-1}\mathbf{x}_i.
\tag{6}
$$

If we assume that $a$ is known. Since $a > 0$, the detection performance of the detector when $a = 1$ is the same as when a takes other values. Thus the MF detector can be denoted as:

$$
\delta^{MF}(\mathbf{x}_i) \;=\; \mathbf{s}^\top \boldsymbol{\Sigma}^{-1}\mathbf{x}_i.
\tag{7}
$$

If $a$ is unknown, under the $H_1$ hypothesis, the maximum likelihood estimate of $a$ is:

$$
a \;=\; \frac{\mathbf{s}^\top \boldsymbol{\Sigma}^{-1}\mathbf{x}_i}{\mathbf{s}^\top \boldsymbol{\Sigma}^{-1}\mathbf{s}}.
\tag{8}
$$

Combining Equations (6) and (8) yields a detector of the following form:

$$\delta^{AMF}(\mathbf{x}_i) = \frac{(\mathbf{s}^\top \boldsymbol{\Sigma}^{-1} \mathbf{x}_i)^2}{\mathbf{s}^\top \boldsymbol{\Sigma}^{-1} \mathbf{s}},\tag{9}$$

which is called Adaptive Matched Filter (AMF) detector [23].

To better establish the connection between the various methods, here we interpret MF from the perspective of data whitening and derive more variant MF detectors. Data whitening is a way to eliminate redundant information from the data. Chang [24] et al., utilized the second-order data statistic of HSI to whiten the data. Specifically, the second-order data statistic is used to characterize the background, and the second-order data statistic is decorrelated to remove the background interference. In this case, the covariance matrix-based whitening is called K-whitening. After K-whitening, $\mathbf{x}_i$ can be denoted as:

$$\tilde{\mathbf{x}}_i = \boldsymbol{\Sigma}^{-1/2} \mathbf{x}_i,\tag{10}$$

It is worth noting that the MF detector as shown in Equation (6) can be viewed as a multiplication of $\mathbf{s}$ and $\mathbf{x}_i$ using K-whitening, respectively, i.e.,

$$\delta^{MF}(\mathbf{x}_i) = a(\boldsymbol{\Sigma}^{-1/2}\mathbf{s})^\top (\boldsymbol{\Sigma}^{-1/2}\mathbf{x}_i) = a\tilde{\mathbf{s}}^\top \tilde{\mathbf{x}}_i.\tag{11}$$

Then setting $a = 1$ and using L2 normalization for $\tilde{\mathbf{s}}$ and $\tilde{\mathbf{x}}_i$, respectively:

$$\delta^{NMF}(\mathbf{x}_i) = \frac{\tilde{\mathbf{s}}^\top \tilde{\mathbf{x}}_i}{\|\tilde{\mathbf{s}}\| \|\tilde{\mathbf{x}}_i\|} = \frac{\tilde{\mathbf{s}}^\top \tilde{\mathbf{x}}_i}{(\tilde{\mathbf{s}}^\top \tilde{\mathbf{s}})^{1/2}(\tilde{\mathbf{x}}_i^\top \tilde{\mathbf{x}}_i)^{1/2}}.\tag{12}$$

The target reference spectrum after K-whitening $\tilde{\mathbf{d}}$ is used to approximate the pure target after K-whitening $\tilde{\mathbf{s}}$ in Equation (12), and then the detector is squared to obtain the Adaptive Coherence (Cosine) Estimator (ACE) [25–27], which is denoted as:

$$\delta^{ACE}(\mathbf{x}_i) = \frac{(\tilde{\mathbf{d}}^\top \tilde{\mathbf{x}}_i)^2}{(\tilde{\mathbf{d}}^\top \tilde{\mathbf{d}})(\tilde{\mathbf{x}}_i^\top \tilde{\mathbf{x}}_i)},\tag{13}$$

where $\tilde{\mathbf{d}} = \boldsymbol{\Sigma}^{-1/2}\mathbf{d}$ and $\tilde{\mathbf{x}}_i = \boldsymbol{\Sigma}^{-1/2}\mathbf{x}_i$. In fact, ACE can also be derived from the spectral angle-based methods, which we will discuss in Section 2.3.

Kraut et al., proposed a detector called Adaptive Subspace Detector (ASD) for target detection, which is originally derived by projecting in subspace and maximizing the signal-to-noise ratio [28]. However, its formal equivalent is to replace the pure target $\mathbf{s}$ with the target reference spectrum $\mathbf{d}$ in the MF detector shown in Equation (6), which is denoted as:

$$\delta^{ASD}(\mathbf{x}_i) = \kappa \mathbf{d}^\top \boldsymbol{\Sigma}^{-1} \mathbf{x}_i,\tag{14}$$

where $\kappa$ is a constant, which normally has little effect on detection.

The hypothesis testing-based detectors represent the target detection problem as a hypothesis testing problem and achieve detection of the target with the help of likelihood ratio theory. In this case, both types of hypothesis introduce the assumption of obeying a Gaussian distribution, and therefore their adaptation to non-Gaussian data is limited.

### 2.3. Spectral Angle-Based Methods

Spectral Angle Mapping (SAM) measures the similarity of spectral properties by calculating the angle between two spectral vectors. The spectral angle between the spectrum of the pixel to be measured $\mathbf{x}_i$ and the reference spectrum of the targets $\mathbf{d}$ is defined as:

$$\cos\theta = \frac{\mathbf{d}^\top \mathbf{x}_i}{\|\mathbf{d}\|\|\mathbf{x}_i\|}.\tag{15}$$

The Spectral Angle Mapping (SAM) detector is obtained by reformulating the spectral angle in the form of a matrix calculation:

$$\delta^{SAM}(\mathbf{x}_i) = \frac{\mathbf{d}^\top \mathbf{x}_i}{(\mathbf{d}^\top \mathbf{d})^{1/2}(\mathbf{x}_i^\top \mathbf{x}_i)^{1/2}}. \tag{16}$$

SAM simply matches the pixel with the reference target spectrum at the pixel level. However, no prior information about the background is considered in the matching process. Therefore, SAM has difficulties suppressing the background interference effectively.

To improve the background suppression performance of SAM, the data whitening technique can be utilized. The reference target spectrum and the pixels in the HSI are K-whitened separately, followed by SAM detection, and finally, the detection results are squared to obtain the ACE [25], as shown in Equation (13). Therefore, ACE can also be seen as a variant of SAM in form. By using K-whitening, the background suppression performance of ACE is improved significantly compared to SAM, as also demonstrated in the experiments in Section 5.1.2. However, SAM and ACE are limited in robustness to spectral variability due to their over-reliance on a given reference target spectrum. Wang et al., introduced the idea of iteratively reweighting to alleviate this problem [26]. Zeng et al., obtained the sparse tensor by 3D tensor decomposition of the original HSI and used SAM to detect the sparse tensor, which effectively suppressed the background information [27].

*2.4. Signal Decomposition-Based Methods*

The signal decomposition-based approach considers the spectrum of each pixel as a combination of different signal components, so that by applying signal decomposition to each pixel, the target can be distinguished from other interference.

For a pixel $\mathbf{x}_i$ to be measured in a hyperspectral image, it can be decomposed into a known signal $\mathbf{t}$ and noise $\mathbf{n}$. The known signal $\mathbf{t}$ can in turn be decomposed into a linear combination of $p$ target spectra $\mathbf{M}\boldsymbol{\alpha}$, where $\mathbf{M} = [\mathbf{m}_1, \mathbf{m}_2, \ldots, \mathbf{m}_p]$ is a matrix consisting of target spectra and $\boldsymbol{\alpha} = [\alpha_1, \alpha_2, \ldots, \alpha_p]^\top$ represents the abundance vector of each target spectrum corresponding to $\mathbf{M}$. Thus, $\mathbf{x}_i$ is denoted as:

$$\mathbf{x}_i = \mathbf{t} + \mathbf{n} = \mathbf{M}\boldsymbol{\alpha} + \mathbf{n}. \tag{17}$$

If there is only one desired target, the remaining $p - 1$ targets are regarded as undesired targets, so $\mathbf{M}$ can in turn be decomposed into a desired target spectral vector $\mathbf{d} = \mathbf{m}_j$ and an undesired target spectral matrix $\mathbf{U} = [\mathbf{m}_1, \mathbf{m}_2, \ldots, \mathbf{m}_{j-1}, \mathbf{m}_{j+1}, \ldots, \mathbf{m}_p]$ . Thus, $\mathbf{x}_i$ is denoted as

$$\mathbf{x}_i = \mathbf{d}\alpha_j + \mathbf{U}\boldsymbol{\gamma} + \mathbf{n}, \tag{18}$$

where $\boldsymbol{\gamma} = [\alpha_1, \alpha_2, \ldots, \alpha_{j-1}, \alpha_{j+1}, \ldots, \alpha_p]^\top$ represents the corresponding abundance vector of $\mathbf{U}$. To eliminate the undesired target matrix $\mathbf{U}$, the following operator is used to project $\mathbf{x}_i$ onto the orthogonal subspace of $\mathbf{U}$:

$$P_\mathbf{U}^\perp = \mathbf{I} - P_\mathbf{U} = \mathbf{I} - \mathbf{U}\mathbf{U}^\#. \tag{19}$$

Thus the Orthogonal Subspace Projection (OSP) detector [29] can be denoted as:

$$\delta^{OSP}(\mathbf{x}_i) = \mathbf{d}^\top P_\mathbf{U}^\perp \mathbf{x}_i. \tag{20}$$

Comparing Equation (7) with Equation (20), OSP is mathematically equivalent to replacing the background elimination method from the inverse of the covariance matrix $\boldsymbol{\Sigma}^{-1}$ of the MF detector to the projection operator $P_\mathbf{U}^\perp$.

OSP detects targets by suppressing undesired targets and enhancing the desired target, and some researchers have developed similar algorithms from this idea. Du et al. proposed signal-decomposed and interference noise (SDIN) based on OSP by considering interfer-



ence characteristics [30]. Chang et al., used Low-Rank and Sparse Matrix Decomposition (LRaSMD) to decompose the pixels to be measured, replacing U with a low-rank matrix and d with a sparse matrix, and introducing the data sphere whitening to further suppress background information [31].

In addition to decomposing the pixel to be measured in the manner described above, Thai et al. proposed to decompose the pixel to be measured into a form of target background and noise, called the signal-background-noise (SBN) model [32]. Unlike OSP, which focuses on improving detection performance by extracting the desired target spectrum, SBN focuses on improving target detection performance by suppressing the background.

### 2.5. Constrained Energy Minimization (CEM)-Based Methods

The basic principle of the constrained energy minimization detector is to design a finite impulse response (FIR) filter that allows only the desired target signature to pass while minimizing the energy output from other signatures [33]. Let the filter coefficient be $\mathbf{w}$. Then the output of the linear filter can be denoted as:

$$y_i = \mathbf{w}^\top \mathbf{x}_i. \tag{21}$$

The number of the pixels in the HSI is $N$, so the average energy of the filter output is

$$\frac{1}{N}\sum_{i=1}^{N} y_i^2 = \mathbf{w}^\top \left[\frac{1}{N}\sum_{i=1}^{N} \mathbf{x}_i \mathbf{x}_i^\top \right]\mathbf{w} = \mathbf{w}^\top \mathbf{R}\mathbf{w}, \tag{22}$$

where $\mathbf{R} = \frac{1}{N}\sum_{i=1}^{N} \mathbf{x}_i \mathbf{x}_i^\top = \frac{1}{N}\mathbf{X}\mathbf{X}^\top$ represents the correlation matrix of the HSI. Minimizing the average energy of the filter output while subject to the constraint $\mathbf{w}^\top \mathbf{d} = 1$, the optimal coefficients of a CEM detector can be obtained by solving the following optimization problem:

$$\min_{\mathbf{w}} \mathbf{w}^\top \mathbf{R}\mathbf{w}$$
$$s.t. \, \mathbf{w}^\top \mathbf{d} = 1. \tag{23}$$

Using the Lagrange multiplier method to solve the above optimization problem, the optimal closed-form solution $\mathbf{w}^{CEM}$ is obtained as:

$$\mathbf{w}^{CEM} = \frac{\mathbf{R}^{-1}\mathbf{d}}{\mathbf{d}^\top \mathbf{R}^{-1}\mathbf{d}}. \tag{24}$$

Thus the CEM detector is:

$$\delta^{CEM}(\mathbf{x}_i) = (\mathbf{w}^{CEM})^\top \mathbf{x}_i = \frac{\mathbf{d}^\top \mathbf{R}^{-1}\mathbf{x}_i}{\mathbf{d}^\top \mathbf{R}^{-1}\mathbf{d}}. \tag{25}$$

We can also explain the CEM detector from another perspective. In addition to K-whitening mentioned in Section 2.2, there is another form of data whitening known as R-whitening. R-whitening utilizes the correlation matrix $\mathbf{R}$ to eliminate background information and is denoted as:

$$\bar{x}_i = \mathbf{R}^{-1/2}\mathbf{x}_i. \tag{26}$$

Therefore the CEM detector can also be derived from the R-whitening combined with the MF detector. Firstly, the whitening method in Equation (11) is replaced by R-whitening from K-whitening and approximating the pure target $s$ with the target reference spectrum $d$, i.e.,

$$\delta^{R-MF}(\mathbf{x}_i) = a\bar{\mathbf{d}}^\top \bar{\mathbf{x}}_i = a(\mathbf{R}^{-1/2}\mathbf{d})^\top (\mathbf{R}^{-1/2}\mathbf{x}_i) = a\mathbf{d}^\top \mathbf{R}^{-1}\mathbf{x}_i. \tag{27}$$

Assuming that the parameter $a$ is unknown, the parameter $a$ is determined by the constraint $\delta^{R-MF}(\mathbf{d}) = 1$:

$$\delta^{R-MF}(\mathbf{d}) = 1 \Rightarrow a = (\mathbf{d}^\top \mathbf{R}^{-1}\mathbf{d})^{-1}. \tag{28}$$

Substituting $a$, the detector is denoted as:

$$\delta^{R-MF}(\mathbf{x}_i) = \frac{\mathbf{d}^\top \mathbf{R}^{-1} \mathbf{x}_i}{\mathbf{d}^\top \mathbf{R}^{-1} \mathbf{d}} = (\mathbf{w}^{CEM})^\top \mathbf{x}_i = \delta^{CEM}(\mathbf{x}_i). \tag{29}$$

Therefore, the CEM detector can also be considered as the MF detector with R-whitening and adaptive parameters.

Chang et al., extended the CEM detector by applying it to detect multiple targets, that is, a matrix composed of desired target spectral vectors as a constrained matrix, and minimizing the output of vectors in other undesired directions, and proposed the linear constrained minimum variance (LCMV) detector [34]. Let the desired target spectral matrix be $\mathbf{D} = [\mathbf{d}_1, \mathbf{d}_2, \ldots, \mathbf{d}_p]$, the optimization problem can be denoted as:

$$\min_{\mathbf{w}} \mathbf{w}^\top \mathbf{R} \mathbf{w}$$
$$s.t. \mathbf{D}^\top \mathbf{w} = \mathbf{c}, \tag{30}$$

where $\mathbf{c} = [c_1, c_2, \ldots, c_p]^\top$ is the constraint vector. Then the closed-form solution is:

$$\mathbf{w}^{LCMV} = \mathbf{R}^{-1} \mathbf{D} (\mathbf{D}^\top \mathbf{R}^{-1} \mathbf{D}) \mathbf{c}. \tag{31}$$

Thus the LCMV detector is:

$$\delta^{LCMV}(\mathbf{x}_i) = (\mathbf{w}^{LCMV})^\top \mathbf{x}_i = \mathbf{x}_i^\top \mathbf{w}^{LCMV} = \mathbf{x}_i^\top \mathbf{R}^{-1} \mathbf{D} (\mathbf{D}^\top \mathbf{R}^{-1} \mathbf{D}) \mathbf{c}. \tag{32}$$

The CEM detector can be viewed as an LCMV detector where the desired target spectral matrix degenerates to the desired target spectral vector and the constraint degenerates to 1, that is, $\mathbf{D} = \mathbf{d}$ and $\mathbf{c} = 1$.

In the real world, the same material will exhibit different spectral properties due to different spatial and temporal factors. Some researchers have eliminated the spectral variations by processing the input to the CEM detector. Zhang et al., proposed a Bayesian Constrained Energy Minimization method (B-CEM) to infer the posterior distribution of the true target spectrum from a given reference target spectrum [35].

Other researchers have borrowed the optimization-based idea of CEM to develop more robust target detection algorithms by modifying the objective function and constraints. RHMF builds the objective function using high-order statistics with a spherical constraint [36]. RMF uses high-order statistics to build the objective function and uses a regularized term [37]. DFMF uses a difference-measured function to build the objective function and utilizes the gradient descent method to find an optimal projection vector [38]. Shi et al. proposed a hyperspectral target detection algorithm that utilizes an inequality constraint to guarantee that the outputs of target spectra, which vary in a certain set, are larger than one [39].

The CEM detector is one type of linear detector, however, real hyperspectral images often contain numerous nonlinear features, so it is critical to improve the non-linear detection capability of the CEM detector. There are currently two main types of methods to improve the non-linear detection performance of CEM detectors. The first type of method directly extends the CEM detector, as shown in Equation (25), from a linear to a nonlinear form. Zou et al., enhance the nonlinear detection by extending the constraints from linear to quadratic [40]. Yang et al., extend the CEM to a more generalized nonlinear form using the deep neural network rather than the FIR filter as the detector [41]. The second type of method improves the nonlinear detection performance by combining multiple CEM detectors. Zou et al., cascade the CEM detectors and suppress the background information with the nonlinear function for the output of each layer to obtain the hierarchical CEM (hCEM) detector [42]. Zhao et al., propose the Ensemble-based Constrained Energy Minimization (E-CEM), which integrates the results of the detection of multiple CEM detectors with different parameters with the help of ensemble learning techniques [43].

Because of its brief form, ease of use, and high reliability, the CEM method is often used in combination with other methods to enhance target detection performance. Ren et al., combined OSP and CEM to propose the Target Constrained Interference Minimization Filter (TCIMF) to reduce the effect of interference signals on detection [44]. Gao et al., combined the Reed-Xiaoli (RX) anomaly detector with the CEM detector to improve the detection performance in complex background situations [45]. In addition, the CEM detector has also been used in the coarse detection stage of some deep learning-based target detection methods [46,47].

*2.6. Kernel-Based Methods*

The kernel-based method can map the data to a high-dimensional space, where higher-order information is used to detect the target, and thus the kernel-based method can better explore the non-linear correlation between spectral bands compared to classical detection methods. Consider the mapping $\mathbf{x}_i \rightarrow \Phi(\mathbf{x}_i)$, and $\Phi(\mathbf{x}_i)$ denotes the high-dimensional pixel, the kernel can be defined as:

$$K(\mathbf{x}_i, \mathbf{x}_j) = \Phi(\mathbf{x}_i)^\top \Phi(\mathbf{x}_j). \tag{33}$$

Then the kernel matrix can be defined as:

$$\mathbf{K} = \Phi(\mathbf{X})^\top \Phi(\mathbf{X}). \tag{34}$$

This definition represents the inner product of the projection of two data samples in a high-dimensional space so that when using the kernel-based method, it is sufficient to replace the inner product in the original detector expression with Equation (33). Most of the classical algorithms discussed previously have been extended to the kernel version, such as KMF [48], KASD [49], KSAM [50], KOSP [51], KCEM [52,53], KTCIMF [54], etc.

Taking KCEM as an example, after mapping to the kernel function space, the CEM detector shown in Equation (25) becomes:

$$\delta^{KCEM}(\Phi(\mathbf{x}_i)) = \frac{\Phi(\mathbf{d})^\top \Phi(\mathbf{R})^{-1} \Phi(\mathbf{x}_i)}{\Phi(\mathbf{d})^\top \Phi(\mathbf{R})^{-1} \Phi(\mathbf{d})}, \tag{35}$$

where $\Phi(\mathbf{R}) = \frac{1}{N}\Phi(\mathbf{X})\Phi(\mathbf{X})^\top = \frac{1}{N}\sum_{i=1}^{N}\Phi(\mathbf{x}_i)\Phi(\mathbf{x}_i)^\top$.

Then, due to the high dimensionality of Equation (35), computing Equation (35) is almost impossible. Therefore, we can borrow the strategy, which is similar to that in KPCA [55]. Let the *j*th eigenvalue of $\Phi(\mathbf{R})$ be $\lambda_j$ and the eigenvector corresponding to $\lambda_j$ be $\mathbf{v}_j$, then there is

$$\Phi(\mathbf{R})\mathbf{v}_j = \lambda_j \mathbf{v}_j = \frac{1}{N}\sum_{i=1}^{N}\Phi(\mathbf{x}_i)\Phi(\mathbf{x}_i)^\top \mathbf{v}_j. \tag{36}$$

Since $\Phi(\mathbf{x}_i)^\top \mathbf{v}_j$ is a scalar, $\mathbf{v}_j$ can be denoted as

$$\mathbf{v}_j = \frac{1}{N\lambda_j}\sum_{i=1}^{N}\Phi(\mathbf{x}_i)\Phi(\mathbf{x}_i)^\top \mathbf{v}_j = \sum_{i=1}^{N}\alpha_j^i \Phi(\mathbf{x}_i) = \Phi(\mathbf{X})\boldsymbol{\alpha}_j, \tag{37}$$

where $\boldsymbol{\alpha}_j = [\alpha_j^1, \alpha_j^2, \ldots, \alpha_j^N]^\top$ is a column vector. Then, multiplying both sides of Equation (36) by $\Phi(\mathbf{X})$ derives

$$\Phi(\mathbf{X})^\top \Phi(\mathbf{R})\mathbf{v}_j = \lambda_j \Phi(\mathbf{X})^\top \mathbf{v}_j. \tag{38}$$

Combining with Equation (37) gives rise to

$$\frac{1}{N}\Phi(\mathbf{X})^\top \Phi(\mathbf{X})\Phi(\mathbf{X})^\top \Phi(\mathbf{X})\boldsymbol{\alpha}_j = \lambda_j \Phi(\mathbf{X})^\top \Phi(\mathbf{X})\boldsymbol{\alpha}_j. \tag{39}$$

Substituting kernel matrix $\mathbf{K}$ into Equation (39), we have:

$$\frac{1}{N}\mathbf{K}^2\boldsymbol{\alpha}_j = \lambda_j\mathbf{K}\boldsymbol{\alpha}_j \Rightarrow \mathbf{K}\boldsymbol{\alpha}_j = N\lambda_j\boldsymbol{\alpha}_j. \tag{40}$$

Therefore, $\boldsymbol{\alpha}_j$ is the eigenvector of $\mathbf{K}$ associated with eigenvalue $N\lambda_j$. $\boldsymbol{\alpha}_j$ can be normalized by $\sqrt{\lambda_j}$ to let $\tilde{\boldsymbol{\alpha}}_j = \boldsymbol{\alpha}_j\sqrt{\lambda_j}$ and $\|\tilde{\boldsymbol{\alpha}}_j\| = 1$ [56]. Then $\mathbf{K}$ can be denoted as

$$\mathbf{K} = N\tilde{\mathbf{A}}\boldsymbol{\Lambda}\tilde{\mathbf{A}}^\top, \tag{41}$$

where $\tilde{\mathbf{A}} = [\tilde{\boldsymbol{\alpha}}_1, \tilde{\boldsymbol{\alpha}}_2, \ldots, \tilde{\boldsymbol{\alpha}}_N]$ is the eigenvector matrix, using $\boldsymbol{\Lambda}^{1/2}$ to normalize $\mathbf{A} = [\boldsymbol{\alpha}_1, \boldsymbol{\alpha}_2, \ldots, \boldsymbol{\alpha}_N]$.

Therefore, $\Phi(\mathbf{R})^{-1}$ can be derived as:

$$\begin{aligned}
\Phi(\mathbf{R})^{-1} &= \mathbf{V}\boldsymbol{\Lambda}^{-1}\mathbf{V}^\top = \Phi(\mathbf{X})\mathbf{A}\boldsymbol{\Lambda}^{-1}\mathbf{A}^\top\Phi(\mathbf{X})^\top \\
&= \Phi(\mathbf{X})\mathbf{A}\boldsymbol{\Lambda}^{1/2}\boldsymbol{\Lambda}^{-2}\boldsymbol{\Lambda}^{1/2}\mathbf{A}^\top\Phi(\mathbf{X})^\top \\
&= N^2\Phi(\mathbf{X})\frac{1}{N^2}\tilde{\mathbf{A}}\boldsymbol{\Lambda}^{-2}\tilde{\mathbf{A}}^\top\Phi(\mathbf{X})^\top \\
&= N^2\Phi(\mathbf{X})\mathbf{K}^{-2}\Phi(\mathbf{X})^\top.
\end{aligned} \tag{42}$$

Substituting Equation (42) into Equation (35), the KCEM detector can be derived as:

$$\begin{aligned}
\delta^{KCEM}(\Phi(\mathbf{x}_i)) &= \Phi(\mathbf{d})^\top\Phi(\mathbf{R})^{-1}\Phi(\mathbf{x}_i)(\Phi(\mathbf{d})^\top\Phi(\mathbf{R})^{-1}\Phi(\mathbf{d}))^{-1} \\
&= \Phi(\mathbf{d})^\top\Phi(\mathbf{X})\mathbf{K}^{-2}\Phi(\mathbf{X})^\top\Phi(\mathbf{x}_i)(\Phi(\mathbf{d})^\top\Phi(\mathbf{X})\mathbf{K}^{-2}\Phi(\mathbf{X})^\top\Phi(\mathbf{d}))^{-1} \\
&= \mathbf{k}_i\mathbf{K}^{-2}\mathbf{k}_d(\mathbf{k}_d\mathbf{K}^{-2}\mathbf{k}_d)^{-1},
\end{aligned} \tag{43}$$

where $\mathbf{k}_i = [K(\mathbf{x}_i, \mathbf{x}_1), K(\mathbf{x}_i, \mathbf{x}_2), \ldots, K(\mathbf{x}_i, \mathbf{x}_N)]^\top$ is a column vector consisting of the kernels between pixel $\mathbf{x}_i$ and each pixel in $\mathbf{X}$ and $\mathbf{k}_d = [K(\mathbf{d}, \mathbf{x}_1), K(\mathbf{d}, \mathbf{x}_2), \ldots, K(\mathbf{d}, \mathbf{x}_N)]^\top$ is a column vector including the kernels between pixel $\mathbf{d}$ and each pixel in $\mathbf{X}$.

The kernel-based methods can better handle nonlinear problems, so they generally have better detection performance than the classical algorithms. However, kernel-based methods also suffer from excessive computational overhead, so a series of approaches, such as Nyström [53,57], has been derived to speed up the computation.

### 2.7. Sparse Representation-Based Methods

Sparse representation-based methods use a linear combination of elements in the dictionary to reconstruct the pixels in the HSI and then use the reconstructed pixels for detection. Therefore, the pixel $\mathbf{x}_i$ can be expressed as

$$\mathbf{x}_i = \mathbf{A}_b\boldsymbol{\alpha}' + \mathbf{A}_t\boldsymbol{\beta}' = [\mathbf{A}_b \ \mathbf{A}_t]\begin{bmatrix}\boldsymbol{\alpha}' \\ \boldsymbol{\beta}'\end{bmatrix} = \mathbf{A}\boldsymbol{\gamma}', \tag{44}$$

where $\mathbf{A}_b$ is an $L \times N_b$ dimensional background dictionary consisting of background training samples (also called atoms), $\boldsymbol{\alpha}'$ is the abundance vector of $\mathbf{A}_b$ corresponding to atoms, $\mathbf{A}_t$ is an $L \times N_t$ dimensional target dictionary consisting of target training samples, and $\boldsymbol{\beta}'$ is the abundance of $\mathbf{A}_t$ corresponding to atom [58].

Given a dictionary $\mathbf{A}$, the reconstructed sparse vector for the pixel $\mathbf{x}_i$ can be obtained by solving the following optimization problem:

$$\begin{aligned}
&\min_{\boldsymbol{\gamma}'} \|\mathbf{A}\boldsymbol{\gamma}' - \mathbf{x}_i\|_2 \\
&s.t. \ \|\boldsymbol{\gamma}'\|_0 \leq K_0,
\end{aligned} \tag{45}$$

where $\|\cdot\|_0$ denotes $l_0$-norm, which is defined as the number of nonzero entries in the vector (or called the sparsity level of the vector) and $K_0$ is a given upper bound on the

sparsity level [59]. Solving the above optimization problem with the greedy algorithm leads to an optimal closed-form solution of the reconstructed sparse vector $\hat{\gamma} = \begin{bmatrix} \hat{\alpha} \\ \hat{\beta} \end{bmatrix}$.

Applying the above reconstructed sparse vectors to target detection, the detector is obtained as:

$$\delta^{STD}(\mathbf{x}_i) = \|\mathbf{x}_i - \mathbf{A}_b\hat{\alpha}\|_2 - \|\mathbf{x}_i - \mathbf{A}_t\hat{\beta}\|_2 \qquad (46)$$

If $\delta^{STD}(\mathbf{x}_i) < \tau'$ with $\tau'$ being a prescribed threshold, then $\mathbf{x}_i$ is determined as a target pixel; otherwise, $\mathbf{x}_i$ is labeled as the background. The above method is called sparse representation for target detection (STD).

Sparse representation-based methods do not require explicit assumptions about the characteristics of the statistical distribution of the observed data, and by selecting the elements of the dictionary appropriately, the algorithm is more robust to spectral variations and more flexible [58,60]. However, sparse representation-based methods rely heavily on the construction of dictionaries, which introduces potential instability. Therefore, it is critical to mitigate this potential instability.

One idea is to prevent model overfitting by adding constraints to the optimization problem as shown in Equation (45). Huang et al., introduce non-local similarity [61,62] to preserve the manifold structure of the original HSI in the sparse representation [63]. Zhang et al., proposed SASTD to improve the detection performance of heterogeneous areas by adaptively constructing a sparse representation model for each pixel by assigning different weights to each neighboring pixel [64]. Huang et al., regularize the original sparse representation model based on the convex relaxation [65] technique to get rid of the problem that the solver may be trapped into a local optimum [66].

Another idea is to make a more accurate and compact representation of the target and background dictionaries. Zhang et al., proposed SRBBH to construct a more reasonable dictionary based on the binary assumption [67]. Wang et al., used spectral angle to select background samples and trained the background dictionary using K singular value decomposition (K-SVD) [68]. Guo et al., combined superpixel segmentation, discriminative structural incoherence, and adaptive embeddable features learning to construct more meaningful target and background dictionaries [69].

In addition, sparsity is one of the inherent properties of hyperspectral data, so combining sparse representation theory with other detection methods can effectively improve the performance of target detection. Yang et al., introduced the sparsity of target pixels into the CEM and the ACE detector and proposed SparseCEM and SparseACE [70]. Li et al., combined sparse representation and collaborative representation to propose the CSCR algorithm, which offers robust detection performance for HSI [71]. In recent years, some methods integrating sparse representation theory and low-rank representation theory have also been successfully applied to hyperspectral target detection [69,72].

### 2.8. Deep Learning-Based Methods

Due to the strong ability to extract nonlinear features and learn underlying distributions, deep learning has been widely used for classification and feature extraction in hyperspectral images over the past decade [73–76]. In recent years, there has been an application of deep learning techniques in hyperspectral target detection [77,78].

Deep learning-based hyperspectral image target detection methods can be divided into two categories based on whether the pipeline used for hyperspectral target detection is end-to-end: one is "end-to-end detection", i.e., to directly use the deep neural networks to determine whether each pixel is the target or the background, and the other one is "detection by reconstruction", i.e., to use the deep learning model to first reconstruct the original HSI and then perform target detection on the reconstructed HSI.

The advantage of "end-to-end detection" is that it can be optimized end-to-end with low complexity, but the disadvantage is that it requires massive data to train the model. "Detection by reconstruction" can obtain more essential features of the feature through reconstruction, which reduces the complexity of subsequent detection tasks and simplifies

the design of detectors. However, it has the disadvantage that the optimization target is not straightforward and the choice of reconstruction model has a large impact on the detection results.

### 2.8.1. End-to-End Detection

Early research has focused on direct detection using deep learning models. One idea is to frame hyperspectral image target detection as a deep learning-based binary classification problem, by setting the target pixels as positive samples and the background pixels as negative samples. This allows target detection by training a neural network model for classification to distinguish the target from the background. Du et al. use convolutional neural networks (CNNs) to determine the class to which each pixel belongs [79]. Freitas et al., introduce 3D convolution for target detection by considering spatial information based on CNNs [77]. Qin et al. use Vision Transformer (ViT) [80] to learn global spectral-spatial features of HSI for target detection [81].

Another idea is to set a pair of pixels belonging to the same target or background as positive samples, and a pair of pixels belonging to the target and sample respectively as negative samples, and train a neural network model to determine whether the input pairs of pixels belong to the same class. For example, CNND [82], TCSNTD [83], and HTD-Net [84] employ this idea. Although the neural network model in such cases is still essentially a neural network for classification, this idea can fully leverage the known spectral prior to transform the target detection problem into a similarity matching problem between the pixel spectra to be measured and the known spectral prior.

However, both types of direct detection methods have some limitations in that the number of samples used for training is insufficient and the positive and negative samples are unbalanced. Therefore, most of the research on direct detection has focused on how to overcome these problems.

One idea is to construct new sample data based on existing sample data. Du et al. obtained the new data by simply subtracting the target and background [79]. With the development of sparse representation models, Zhu et al. generated background samples with the help of sparse representation methods and mixed the background samples with the target before generating target samples [83]. Generative models based on deep learning have recently made great development, and these models have proven to be beneficial for training downstream tasks. As a result, researchers have started investigating approaches to generate new hyperspectral data using these models.

For example, Zhang et al., used autoencoder (AE) to generate target samples and then use the linear prediction (LP) strategy to find background samples [84] while Gao et al., relies on generative adversarial networks (GAN) [85] to generate additional target and background samples [86].

Another idea is to explore methods that do not require excessive sample data, so some researchers have turned their attention to few-shot learning. Few-shot learning refers to learning from a small number of labeled samples [87], and Siamese Network is one of the main approaches used to solving the problem of few-shot learning [88]. Siamese Network connects a pair of neural networks with shared weights at the output and learns a function that can measure the similarity between two samples [89]. In recent years, many methods based on Siamese networks have been proposed, such as LRS-STD [90], Siamese fully connected target detector (SFCTD) [91], Siamese transformer target detector (STTD) [92], and meta learning-based Siamese network (MLSN) [93].

### 2.8.2. Detection by Reconstruction

HSI contains interference information that is not conducive to target detection. By reconstructing the original HSI, a new representation that better reflects the characteristics of the original HSI features can be obtained. Initially, researchers reconstructed HSI by traditional methods such as band selection and then used detectors to perform detection to eliminate redundant information and enhance useful information, thus improving de-

tection performance [94]. In recent years, generative models based on deep learning have made great development, and some generative models such as autoencoder (AE), variational autoencoder (VAE) [95], and generative adversarial networks (GAN) can obtain the essential features in the original HSI compared with the traditional reconstruction methods, and reconstruct the original HSI into a more convenient feature space for detection.

The most intuitive idea is to reconstruct HSI directly using generative models based on deep learning. One of the most representative models is the autoencoder (AE). AE is a self-supervised model consisting of an encoder and decoder that learns the essential features of the input data and can remove noise and redundant information by reconstructing the input data. Therefore, Shi et al. proposed DCSSAED [96] and 3DMMRAE [97] methods by reconstructing hyperspectral images with the help of the AE model. The DCSSAED method adds the constraint of maximizing the distance between the background and target in the feature space during the training of SSAE, after which the reconstructed image is sent to the RBF detector for detection. In the 3DMMRAE method, in contrast, AE is combined with 3D convolution to introduce spatial information into the reconstruction to generate a more complete representation, after which the hRBF detector is used for detection.

Based on AE, Kingma et al., proposed the variational autoencoder (VAE), which converts the prediction of latent variables into the prediction of the distribution parameters of latent variables. VAE has more advantages than AE in terms of generative ability, continuity in latent space, interpretability, and training stability. Xie et al., reconstructed HSI by utilizing the VAE model. After reconstruction, they weighted the feature maps and then used morphological methods for detection [98].

Generative adversarial networks (GANs) reconstruct images by adversarial learning between generators and discriminators, without minimizing reconstruction loss but using discriminators to guide image reconstruction, achieving better reconstruction quality. Xie et al., applied GANs to hyperspectral image reconstruction, using adversarial learning to ensure the validity of the latent features extracted by the network, and the reconstructed images are detected in the spatial dimension and spectral dimension, respectively [99].

In addition to directly utilizing generative models, some researchers have also combined traditional reconstruction methods with generative models to obtain better reconstruction results. For example, Xie et al., followed the VAE with band selection to obtain a more detection-friendly representation of the original HSI [100]. Since the ultimate goal of reconstruction is to facilitate target detection, the coarse detection results of the conventional detector can be used to guide the reconstruction. Xie et al. use the coarse detection results of the CEM detector to select the background, and later reconstruct the background with the encoder-decoder structure [46]. Shi et al., use the CEM detector and Gaussian filter to obtain the Region of Interest (RoI), and then use the RoI as the model input to reconstruct the image [47].

Recently, methods based on deep metric learning [101,102] and contrastive learning [103–105] have also been migrated to the hyperspectral target detection task. The ultimate goal of both deep metric learning and contrastive learning is to draw similar samples closer and push away dissimilar samples in the feature space, but deep metric learning is generally supervised, while contrastive learning is usually self-supervised. For metric learning, Zhu et al., use a deep metric network to reconstruct the target and background samples in the feature space and determine the target by computing the distance between the pixel to be measured and the target reference spectrum in the feature space [106]. For contrastive learning, Wang et al., treat the augmented samples from the same pixel as positive samples and the augmented samples from different pixels as negative samples, so the target detection is transformed into the matching of the pixel to be measured and the target reference spectrum in the feature space [107].

## 3. Summary and Comparison

In Figure 1, we build a network that connects the algorithms mentioned in Section 2 based on the representative algorithms in each class.

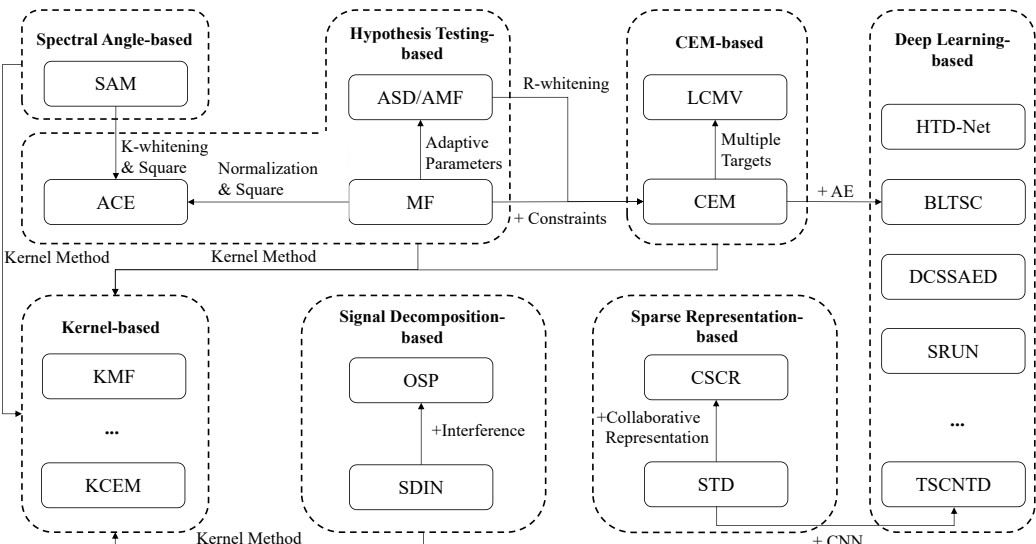

**Figure 1.** The relationship between the target detection algorithms.

According to Figure 1, we start from the SAM as shown in Equation (16). Using the K-whitening as shown in Equation (10) for SAM and squaring it, we can obtain ACE as shown in Equation (13). In addition, we can start from the MF as shown in Equation (11) and perform normalization and K-whitening on it, which also yields ACE. Thus, the connection between the spectral angle-based method and the hypothesis testing-based method is established.

The hypothesis test-based method is unconstrained, and the CEM shown in Equation (25) can be obtained by adding constraints to MF and minimizing its output energy. At the same time, the CEM method can also be seen as adding adaptive parameters shown in Equation (8) to the MF and using R-whitening shown in Equation (26). Thus, the connection between the hypothesis testing-based and CEM-based method is established.

Furthermore, it is observed that there is some formal similarity between the MF shown in Equation (7) and the OSP shown in Equation (20). Thus, a connection between the signal decomposition-based and hypothesis testing-based method is established.

The four classes of methods mentioned above can all be extended to the kernel version according to Equation (33), so this establishes the connection between the kernel-based methods and the four classes of methods mentioned above.

In recent years, deep learning-based methods are known as a hot research topic due to their high accuracy. Deep learning-based methods are further divided into two types of solution ideas: end-to-end detection and detection by reconstruction. Among them, sparse representation-based methods can provide training samples for end-to-end detection (e.g., TSCNTD [83]), while traditional methods (e.g., CEM) can provide the prior information for detection by reconstruction (e.g., BLTSC [46]).

In Table 1, we summarize the methods presented in Section 2. We summarize the basic idea and limitation of seven categories of methods, and list some representative algorithms in each category, summarizing the prior input information they need in practical applications.

**Table 1.** Summary of methods.

| Methodology | Basic Idea | Example Algorithms | Input/Required | Limitations |
|---|---|---|---|---|
| Hypothesis testing | Calculating the likelihood ratio under the two hypotheses | MF [22] | target $\mathbf{d}$, HSI $\mathbf{X}$ | Limited performance on non-Gaussian data |
| | | ACE [25] | target $\mathbf{d}$, HSI $\mathbf{X}$ | |
| | | ASD [28] | target $\mathbf{d}$, HSI $\mathbf{X}$ | |
| Spectral angle | Calculating the cosine similarity between two spectral vectors | SAM | target $\mathbf{d}$, HSI $\mathbf{X}$ | Limited robustness to spectral variations |
| Signal decomposition | Decomposing the signal into subspaces according to certain rules | OSP [29] | target $\mathbf{d}$, undesired target matrix $\mathbf{U}$, HSI $\mathbf{X}$ | Too much input information required |
| | | SDIN [30] | target $\mathbf{d}$, interference subspace $\mathbf{\Psi}$, HSI $\mathbf{X}$ | |
| | | SBN [32] | target $\mathbf{d}$, background matrix $\mathbf{B}$, HSI $\mathbf{X}$ | |
| CEM-based | Designing the FIR filter that minimizes the output energy and allows only the target to pass | CEM [33] | target $\mathbf{d}$, HSI $\mathbf{X}$ | Limited performance on non-Gaussian data |
| | | LCMV [34] | target matrix $\mathbf{D}$, target constraint vector $\mathbf{c}$, HSI $\mathbf{X}$ | |
| | | TCIMF [44] | target matrix $\mathbf{D}$, undesired target matrix $\mathbf{U}$, HSI $\mathbf{X}$ | |
| | | RHMF [36] | target $\mathbf{d}$, HSI $\mathbf{X}$, tolerance $\epsilon$ ,high-order differentiable function $G(x)$ | |
| | | hCEM [42] | target $\mathbf{d}$, HSI $\mathbf{X}$, tolerance $\delta_k$ | |
| | | ECEM [43] | target $\mathbf{d}$, HSI $\mathbf{X}$, window number $n$, detection layer number $k$, CEM number per layer $m$ | |
| Kernel-based | Mapping the data to a high-dimensional kernel space | KSAM [50] | target $\mathbf{d}$, HSI $\mathbf{X}$, kernel function $\Phi(x)$ | High computation and memory cost |
| | | KMF [48] | target $\mathbf{d}$, HSI $\mathbf{X}$, kernel function $\Phi(x)$ | |
| | | KOSP [51] | target $\mathbf{d}$, undesired target matrix $\mathbf{U}$, HSI $\mathbf{X}$, kernel function $\Phi(x)$ | |
| | | KCEM [52] | target $\mathbf{d}$, HSI $\mathbf{X}$, kernel function $\Phi(x)$ | |
| Sparse representation | Utilizing a linear combination of elements in the dictionary to represent the HSI | STD [58] | dictionary $\mathbf{A}$, HSI $\mathbf{X}$ | Potential instability due to different dictionaries |
| | | CSCR [71] | dictionary $\mathbf{A}$, HSI $\mathbf{X}$, regularization parameter $\lambda_1, \lambda_2$, window size $w_{in}, w_{out}$ | |
| | | SASTD [64] | dictionary $\mathbf{A}$, HSI $\mathbf{X}$, sparsity level $l$, window sizes $w_s, w_w, w_b$ | |
| | | SRBBH [67] | dictionary $\mathbf{A}$, HSI $\mathbf{X}$, sparsity level $l$, dual-window sizes $w_{OWR}, w_{IWR}$ | |

**Table 1.** *Cont.*

| Methodology | Basic Idea | Example Algorithms | Input/Required | Limitations |
|---|---|---|---|---|
| Deep learning | Learning the intrinsic patterns and representation of sample data using neural networks etc. | TSCNTD [83] | target $\mathbf{d}$, HSI $\mathbf{X}$ | Low data availability and limited model transferability |
| | | HTD-Net [84] | target samples $\mathbf{T}$, HSI $\mathbf{X}$ | |
| | | DCSSAED [96] | target samples $\mathbf{T}$, HSI $\mathbf{X}$, adjustable parameter $\sigma_1, \sigma_2$ | |
| | | SRUN [98] | target $\mathbf{d}$, HSI $\mathbf{X}$, parameters depth $d$, number of hidden nodes $h$, regularization parameter $\alpha$, threshold $\tau$ | |
| | | BLTSC [46] | target $\mathbf{d}$, HSI $\mathbf{X}$, normalized initial detection result $\mathbf{D}_1$, parameter $\lambda$ | |
| | | 3DMMRAED [97] | target $\mathbf{d}$, HSI $\mathbf{X}$, number of iteration $i$ | |

## 4. Datasets and Metrics

### 4.1. Datasets

Hyperspectral target detection datasets typically consist of HSIs and ground truth maps, which can be used to evaluate the target detection performance of an algorithm. For some data-driven algorithms, the datasets can also be used as training samples. Some common datasets and their basics are shown in Table 2. Researchers often crop a part of the dataset to perform experiments for target detection.

**Table 2.** Commonly used datasets for target detection.

| Dataset | Sensor | Spatial Size (Pixels) | Spectral Bands | Size of the Part Used for Target Detection (Pixels) | Number of Target Pixels |
|---|---|---|---|---|---|
| Cuprite [98] | AVIRIS | $512 \times 614$ | 224 | $250 \times 191$ | 39 |
| San Diego [99] | AVIRIS | $400 \times 400$ | 224 | $200 \times 200$ | 134 |
| Airport-Beach-Urban [108] | AVIRIS and ROSIS-03 | $100 \times 100$ | 224 | $100 \times 100$ | / |
| HYDICE Urban [96] | HYDICE | $307 \times 307$ | 210 | $80 \times 100$ | 21 |
| HYDICE Forest [84] | HYDICE | $64 \times 64$ | 210 | $100 \times 100$ | 19 |
| Cooke City [109] | HyMap | $280 \times 800$ | 126 | $100 \times 300$ | 118 |

Some of the datasets mentioned in Table 2 are shown in Figure 2, which include the cropped hyperspectral image and the corresponding ground truth map.

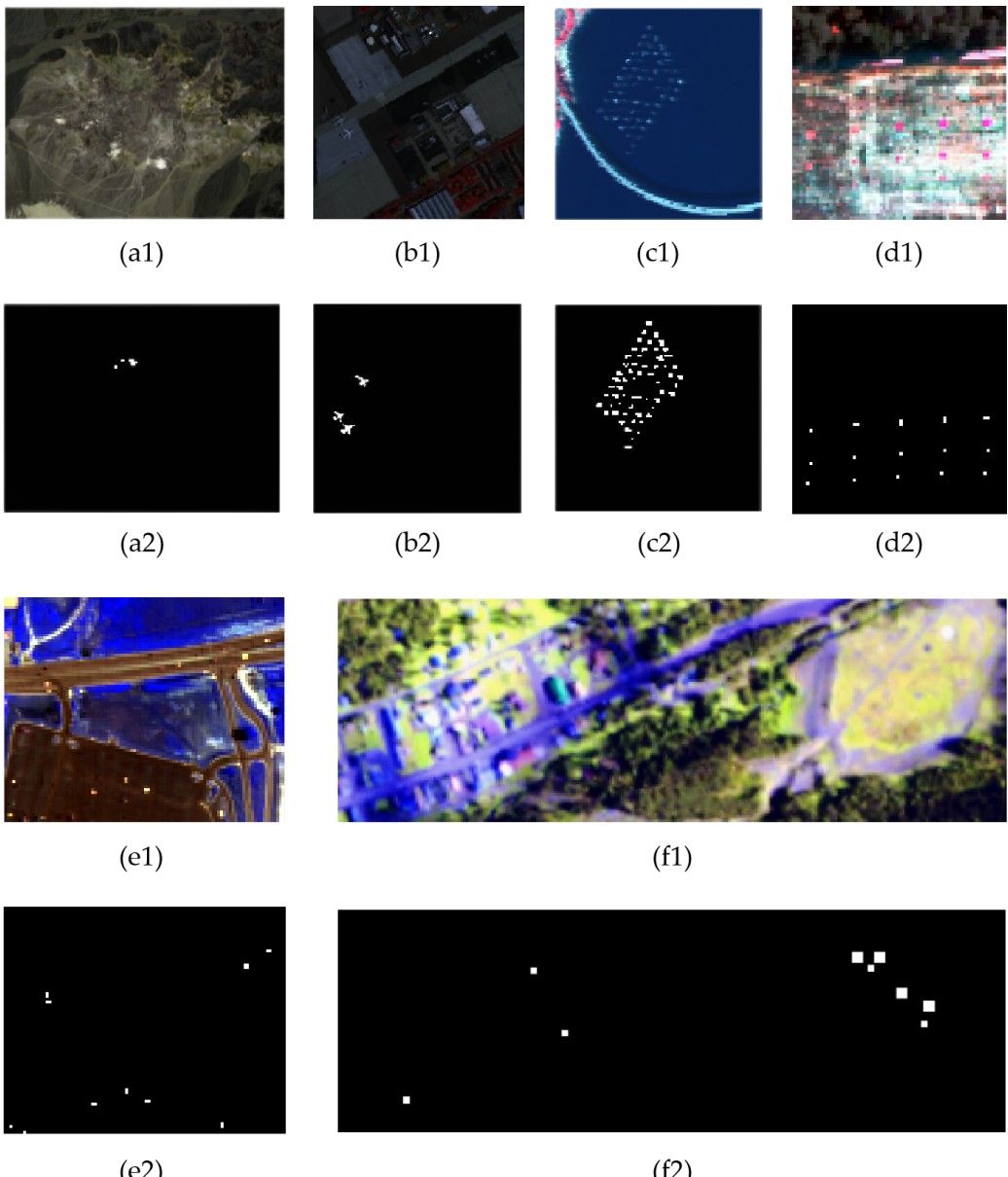

**Figure 2.** Some of the commonly used datasets cropped by researchers, with false-color image and ground truth. (**a**) Cuprite [98]. (**b**) San Diego [99]. (**c**) Airport-Beach-Urban [108]. (**d**) HYDICE Forest [84]. (**e**) HYDICE Urban [96]. (**f**) Cooke City [109]. A suffix of 1 represents the pseudo-color image (like a1), a suffix of 2 represents the ground truth (like a2).

*4.2. Evaluation Metrics*

4.2.1. Receiver Operating Characteristic (ROC) Curve and Area under ROC Curve (AUC)

To evaluate the detection performance of the algorithms, the receiver operating characteristic (ROC) curve and the area under the curve (AUC) was used for quantitative analysis. By changing the threshold $\tau$, we can obtain different detection probability $P_D$ and false alarm probability $P_F$. $P_D$ and $P_F$ can be calculated by the following procedure:

$$P_D(\tau) = \frac{TP}{TP + FN}, \; P_F(\tau) = \frac{FP}{FP + TN}, \tag{47}$$

where *TP* and *FN* denote the number of target pixels correctly detected and the number of pixels that are indeed targets but not detected under the threshold $\tau$, while *FP* and *TN* denote the number of background pixels incorrectly detected as targets and the number of background pixels correctly detected under the threshold $\tau$.

For a given threshold $\tau$, $(P_F(\tau), P_D(\tau))$ can be obtained by Equation (47), which is regarded as a point on the Cartesian coordinate system. By setting different $\tau$, different points on the coordinate system can be obtained. The curve formed by these points is called ROC. The area enclosed by the ROC and $P_F$ axes is called AUC. This area can be mathematically expressed using the integral as follows:

$$AUC = \int_0^1 P_D(\tau) dP_F(\tau). \tag{48}$$

However, in practice, $P_D$ and $P_F$ are not continuous and $P_F$ is not equally spaced, so the trapezoidal rule is generally used to approximate the solution.

### 4.2.2. 3D-ROC

To evaluate the detection performance more precisely, Chang et al. proposed 3D-ROC curves by generating three 2D ROC curves for $(P_D, P_F)$, $(P_D, \tau)$, $(P_F, \tau)$ and the corresponding $AUC_{(P_D,P_F)}$, $AUC_{(P_D,\tau)}$, $AUC_{(P_F,\tau)}$ to evaluate the detector effectiveness, target highlighting ability and background suppression ability, respectively [110]. In addition, based on the three AUCs mentioned above, Chang developed two more indicators for a comprehensive evaluation, calculated as follows:

$$AUC_{OA} = AUC_{(P_D,P_F)} + AUC_{(P_D,\tau)} - AUC_{(P_F,\tau)} \tag{49}$$

$$AUC_{SNPR} = \frac{AUC_{(P_D,\tau)}}{AUC_{(P_F,\tau)}} \tag{50}$$

$AUC_{OA}$ measures the overall performance by weighting three 2D metrics. Since larger $AUC_{(P_D,P_F)}$ and $AUC_{(P_D,\tau)}$ correspond to better performance, their weights are positive, while smaller $AUC_{(P_F,\tau)}$ corresponds to better background suppression performance, hence the negative weights. SNPR draws on the concept of signal-to-noise ratio, where the target is considered as information and the background as noise. The larger $AUC_{OA}$ and $AUC_{SNPR}$, the better performance of the detector.

## 5. Discussion

In this section, we summarize and compare the methods mentioned in this paper and give our views on future research directions.

### 5.1. Experiments

#### 5.1.1. Acquisition of the Target Spectrum

In target detection methods, the reference spectrum of the target is always needed. However, the acquisition method is not consistent in previous papers, which makes it difficult to make fair comparisons of detection performance between methods.

The current methods for selecting target spectrums are mainly three different approaches: (1) averaging the spectra of all target pixels, (2) randomly selecting the spectrum of one or a few pixels among all target pixels as the detection spectrum, and (3) selecting the spectra of the corresponding class in the spectral library. However, all the above methods have some problems. Method (1) is costly to acquire, method (2) may introduce randomness, and the spectra in the spectral library used in method (3) are measured in a library environment, which differs greatly from the real world, and some targets have not been included in the spectral library.

To tackle the above problems, we propose a method that takes both real-world prior information and randomness into account. Assuming that the spectral properties of neighboring target pixels are connected, so the target region can be segmented into $k$ regions, and then one representative pixel from each region is selected as the reference target spectrum, and the target spectrum vectors of the $k$ pixels are averaged if only one reference target spectrum is needed for detection.

Specifically, k-means clustering is performed on the set of location sittings of each target in the ground truth, and the target pixel closest to the center of each cluster is taken as the representative target pixel in each region, and its corresponding spectrum is taken as the reference target spectrum.

### 5.1.2. Experiment Performances

We experiment with some of the algorithms mentioned in Section 2 that are open source in code and evaluate performance on San Diego and Cuprite datasets.

*A. San Diego dataset*

For the San Diego dataset, the size of the cropped image used for the experiment is $200 \times 200$ and the target to be detected is 3 airplanes with a total of 134 pixels. The target reference spectrum is obtained by the criterion mentioned in Section 5.1.1 with the number of cluster centers $k = 3$.

For hCEM, we set the tolerance $\delta_k$ to less than $10^{-6}$. For ECEM, we set the number of windows to $n = 4$, and fix the detection layers and the number of CEMs per layer to $k = 10$ and $m = 6$. For CSCR, the parameters $\lambda_1$, $\lambda_2$ and the window size $(w_{in}, w_{out})$ are fixed to $10^{-1}$, $10^{-2}$ and $(11, 5)$, respectively. For BLTSC, we set the threshold of the binarization of coarse detection result $\epsilon$ and the detection parameter $\lambda$ to 0.15 and 10. We used 3D-ROC analysis and selected five metrics $AUC_{(P_D,P_F)}$, $AUC_{(P_D,\tau)}$, $AUC_{(P_F,\tau)}$, $AUC_{OA}$, and $AUC_{SNPR}$, mentioned in Section 4.2, to comprehensively evaluate the detection performance of the algorithms, as shown in Table 3. The detection results of the algorithms are shown in Figure 3. The 2D ROC curves and 3D-ROC curves of the algorithms are shown in Figure 4.

**Table 3.** 3D-ROC analysis for some algorithms in the San Diego dataset. Bold indicates the best value under the metric, and underline indicates the second-best value under the metric.

| Methodology | Algorithm | $AUC_{(P_D,P_F)}$ | $AUC_{(P_D,\tau)}$ | $AUC_{(P_F,\tau)}$ | $AUC_{SNPR}$ | $AUC_{OA}$ |
|---|---|---|---|---|---|---|
| Hypothesis testing | MF | 0.8969 | 0.4031 | 0.2190 | 1.8405 | 1.0810 |
| | ACE | 0.8955 | 0.1910 | <u>0.0051</u> | <u>37.2919</u> | 1.0814 |
| Spectral angle | SAM | 0.7633 | 0.1969 | 0.0900 | 2.1869 | 0.8701 |
| CEM-based | CEM | 0.8937 | 0.3968 | 0.2103 | 1.8872 | 1.0803 |
| | hCEM | <u>0.9916</u> | 0.5128 | 0.0155 | 33.1421 | <u>1.4890</u> |
| | ECEM | **0.9922** | <u>0.5243</u> | 0.0150 | 34.9697 | **1.5015** |
| Sparse representation model | CSCR | 0.9842 | **0.6060** | 0.4776 | 1.2688 | 1.1126 |
| Deep learning | BLTSC | 0.8999 | 0.1428 | **0.0018** | **80.7670** | 1.0409 |

From Table 3 and Figure 4, CSCR and ECEM have better target highlighting performance. BLTSC and ACE have better background suppression performance on our San Diego dataset. On the three metrics indicating comprehensive detection performance, ECEM and hCEM performed better on the $AUC_{(P_D,P_F)}$ and $AUC_{OA}$ metrics, and BLTSC and ACE performed better on the $AUC_{SNPR}$ metric.

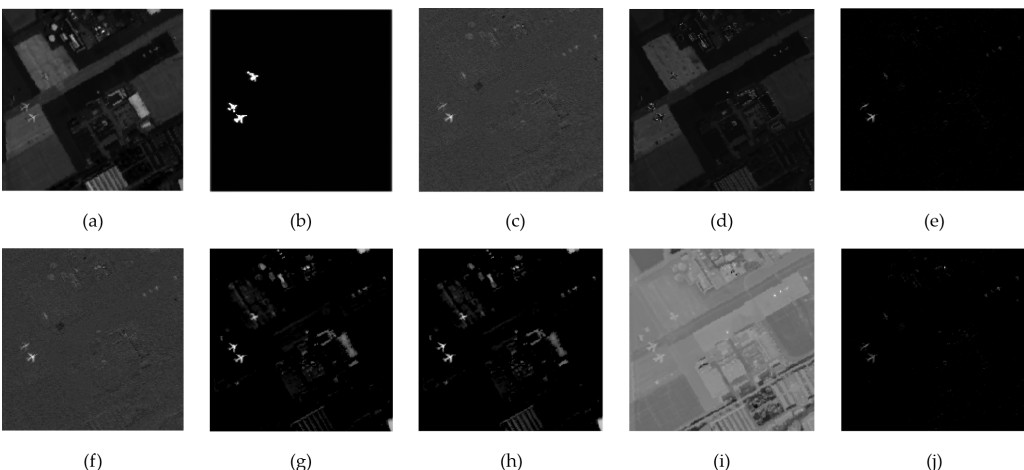

**Figure 3.** The detection results of algorithms on the cropped San Diego dataset. (**a**) The first band of the image. (**b**) ground truth. (**c**) MF. (**d**) SAM. (**e**) ACE. (**f**) CEM. (**g**) hCEM. (**h**) ECEM. (**i**) CSCR. (**j**) BLTSC.

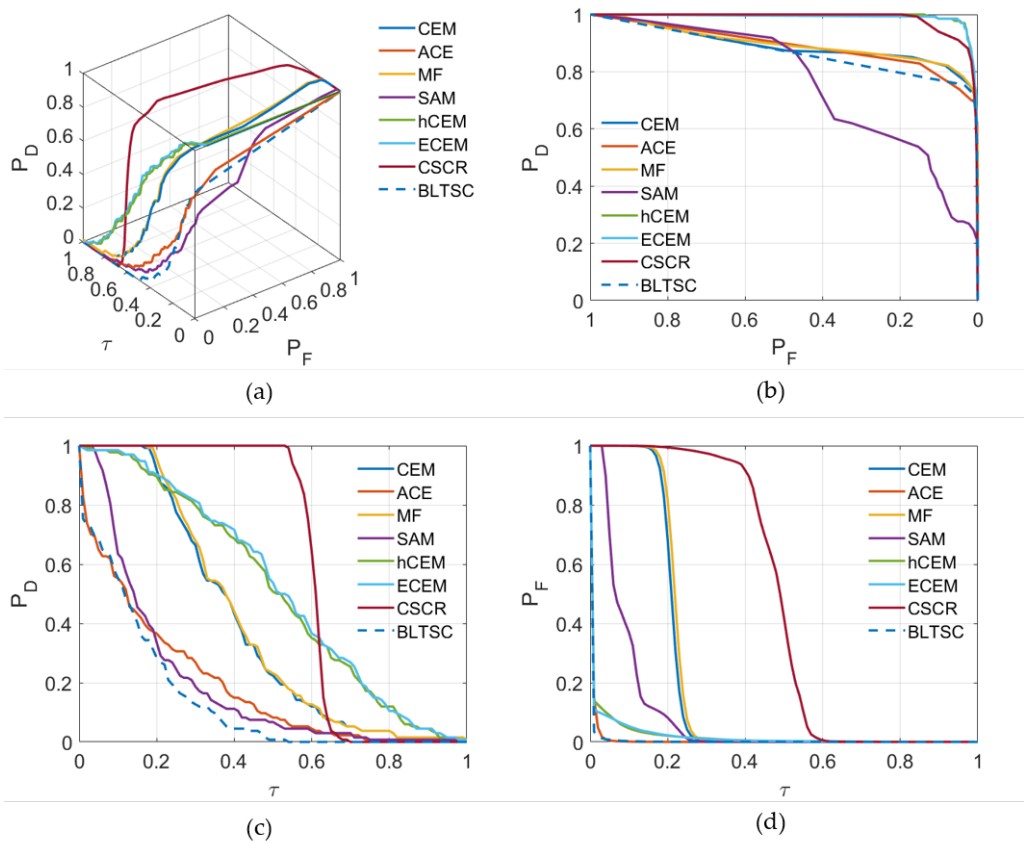

**Figure 4.** 3D-ROC curve on the cropped San Diego dataset along with its three generated 2D ROC curves. (**a**) 3D ROC curve. (**b**) 2D ROC curve of $(P_D, P_F)$. (**c**) 2D ROC curve of $(P_D, \tau)$. (**d**) 2D ROC curve of $(P_F, \tau)$.

*B. Cuprite dataset*

For the Cuprite dataset, the size of the cropped image used for the experiment is $250 \times 191$ and the target to be detected is buddingtonite, which occupies 39 pixels. The target reference spectrum is obtained by the criterion mentioned in Section 5.1.1 with the number of cluster centers $k = 3$.

For hCEM, we set the tolerance $\delta_k$ to less than $10^{-6}$. For ECEM, we set the number of windows to $n = 4$, and fix the detection layers and the number of CEMs per layer to $k = 10$

and $m = 6$. For CSCR, The parameters $\lambda_1$, $\lambda_2$ and the window size $(w_{in}, w_{out})$ are fixed to $10^{-1}$, $10^{-2}$, and $(11, 5)$, respectively. For BLTSC, we set the threshold of the binarization of the coarse detection result $\epsilon$ and the detection parameter $\lambda$ to 0.15 and 8. We used 3D-ROC analysis and selected five metrics $AUC_{(P_D,P_F)}$, $AUC_{(P_D,\tau)}$, $AUC_{(P_F,\tau)}$, $AUC_{OA}$ and $AUC_{SNPR}$ mentioned in Section 4.2 to comprehensively evaluate the detection performance of the algorithms, as shown in Table 4. The detection results of the algorithms are shown in Figure 5. The 2D ROC curves and 3D-ROC curves of the algorithms are shown in Figure 6.

**Table 4.** 3D-ROC analysis for some algorithms in Cuprite dataset. Bold indicates the best value under the metric, and underline indicates the second-best value under the metric.

| Methodology | Algorithm | $AUC_{(P_D,P_F)}$ | $AUC_{(P_D,\tau)}$ | $AUC_{(P_F,\tau)}$ | $AUC_{SNPR}$ | $AUC_{OA}$ |
|---|---|---|---|---|---|---|
| Hypothesis testing | MF | 0.9743 | 0.5050 | 0.2585 | 1.9534 | 1.2208 |
| | ACE | 0.9489 | 0.1825 | **0.0096** | 19.0289 | 1.1218 |
| Spectral angle | SAM | 0.9119 | 0.4146 | 0.1743 | 2.3779 | 1.1522 |
| CEM-based | CEM | 0.9759 | 0.5097 | 0.2573 | 1.9808 | 1.2283 |
| | hCEM | **0.9918** | 0.3984 | 0.0197 | <u>20.2127</u> | <u>1.3705</u> |
| | ECEM | <u>0.9792</u> | <u>0.6534</u> | 0.0805 | 8.1122 | **1.5520** |
| Sparse representation model | CSCR | 0.9709 | **0.8997** | 0.7931 | 1.1344 | 1.0775 |
| Deep learning | BLTSC | 0.9620 | 0.2658 | <u>0.0083</u> | **31.8302** | 1.2194 |

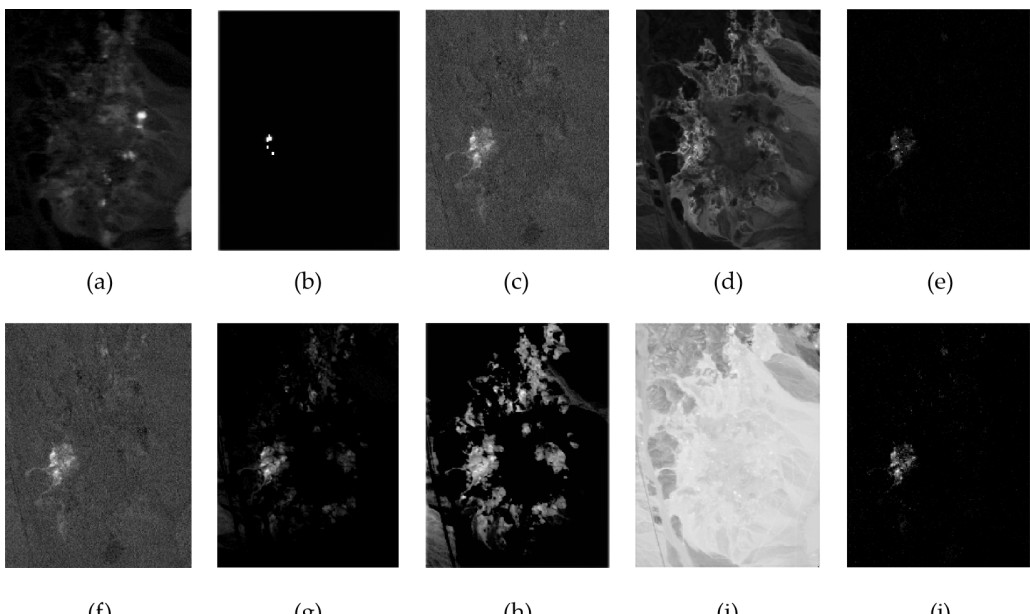

| (a) | (b) | (c) | (d) | (e) |
|---|---|---|---|---|

| (f) | (g) | (h) | (i) | (j) |
|---|---|---|---|---|

**Figure 5.** The detection results of algorithms on the cropped Cuprite dataset. (**a**) The first band of the image. (**b**) ground truth. (**c**) MF. (**d**) SAM. (**e**) ACE. (**f**) CEM. (**g**) hCEM. (**h**) ECEM. (**i**) CSCR. (**j**) BLTSC.

From Table 4 and Figure 6, CSCR and ECEM have better target highlighting performance, while ACE and BLTSC have better background suppression performance on our Cuprite dataset. On the three metrics indicating comprehensive detection performance, ECEM and hCEM performed better on the $AUC_{(P_D,P_F)}$ and $AUC_{OA}$ metrics and BLTSC and hCEM performed better on the $AUC_{SNPR}$ metric.

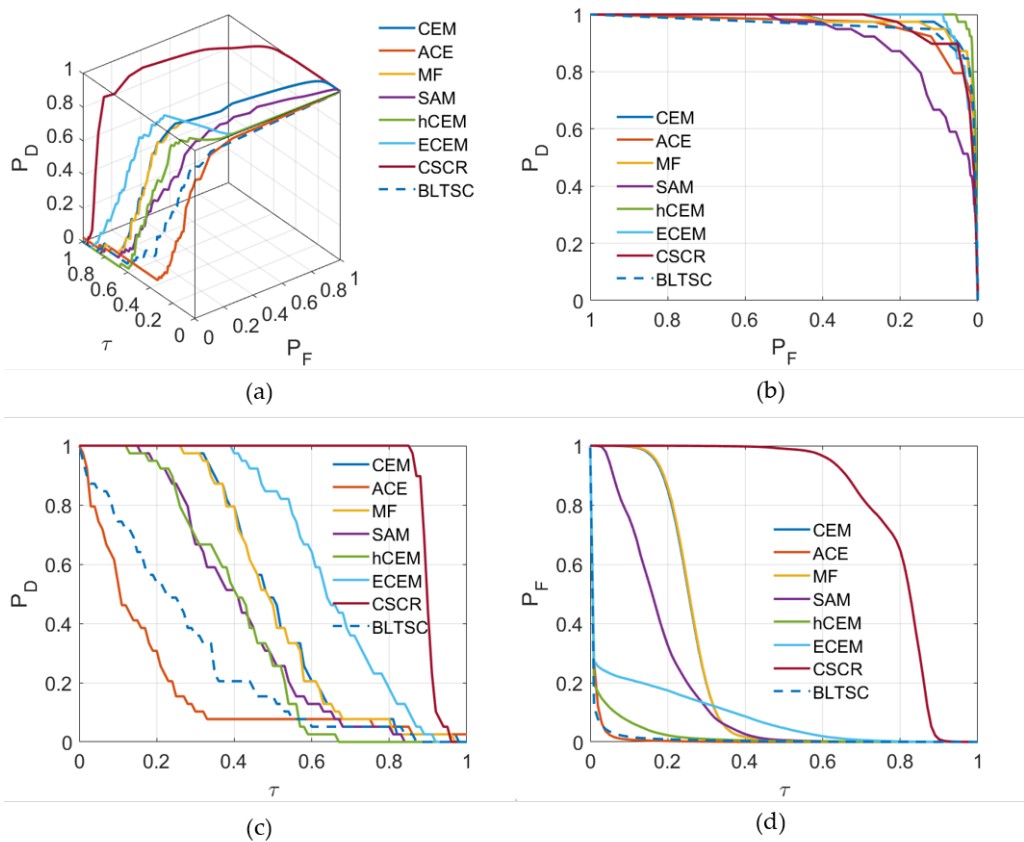

**Figure 6.** 3D-ROC curve on the cropped Cuprite dataset along with its generated three 2D ROC curves. (**a**) 3D ROC curve. (**b**) 2D ROC curve of $(P_D, P_F)$. (**c**) 2D ROC curve of $(P_D, \tau)$. (**d**) 2D ROC curve of $(P_F, \tau)$.

*5.2. Future Challenges*

5.2.1. Spectral Variability

During the acquisition of the HSI, changes in the atmosphere, illumination, environmental conditions, etc., may cause the same ground objects to exhibit different spectral characteristics. In contrast, two different ground objects may also exhibit the same spectral characteristics under certain conditions. In addition, due to the low spatial resolution of the HSI captured by the hyperspectral sensor, each pixel of the target is likely to be a mixture of target and non-target features, and the target, in this case, is called a sub-pixel target [111,112]. The presence of subpixel targets also brings about variations in the spectral features in the real HSI.

Such properties of HSI are known as spectral variability [113], which brings diversity to the spectral signature of the target and makes detection much more difficult. To tackle this problem, future solutions may be attempted in two ways. The first is to increase the number of reference target spectra to better characterize the target by obtaining more diverse samples of the target. The second is to enhance the robustness of the algorithm to spectral variability by mining the essential information in the existing target reference spectra and fully integrating spatial information to overcome the degradation of accuracy.

5.2.2. Acquisition of the Ground Truth

Experimental data with ground truth is difficult to obtain and requires time-consuming and costly fieldwork by professionals. Some recent deep learning models can achieve pixel-level accurate annotation on natural images [114,115]. Therefore, the development of automatic annotation methods adapted to hyperspectral images has become an important way to obtain more experimental data with ground truth.

### 5.2.3. Causal Real-Time Detection

Hyperspectral images are typically acquired by pushbroom or whiskbroom sensing modes, using line-based or pixel-based scanning, respectively [116]. The special imaging mode requires that the information used by the real-time detector must be before the pixel to be measured and cannot use future information that has not yet been acquired [117]. For pushbroom sensing, a full row of hyperspectral images is acquired per scan, so the pixels available for detection are the current row and the row before it; for whiskbroom sensing, a single pixel is acquired per scan, so the pixels available for detection are only all the pixels before the current pixel. We refer to hyperspectral target detectors that follow this particular type of detection pattern as causal real-time detectors [118]. Hyperspectral target detection in causal real-time is important for tasks such as military reconnaissance and disaster relief, but there is a lack of research on related algorithms. In recent years, autoregressive models naturally have causality and have made great progress in the fields of image generation [119–121] and text prediction [122,123], so hardware-friendly and real-time autoregressive methods may be the important way to solve this problem.

### 5.2.4. Challenges in Deep Learning-Based Methods

Deep learning methods can better extract the features and latent information of HSIs and have been shown to have better detection performance in target detection compared to other methods, which is an important research direction for future target detection algorithms. However, deep learning-based methods have some unique problems compared to other methods at present.

First, the unbalanced numbers between different samples. The number of target and background samples contained in hyperspectral images is unbalanced, which can have a significant impact on model performance. For this problem, data augmentation, few-shot learning, and self-supervised learning have proven their effectiveness, and further development of these techniques may be required in the future.

Second, high computational and time overhead. Compared with traditional methods, deep learning-based methods improve detection accuracy but are also accompanied by a surge in the number of parameters and computational complexity. In addition, for new HSIs, deep learning-based methods often need to be re-trained, which consumes a lot of computational and time resources. For this, the generalization ability of the model to different data can be enhanced by improving the model design or increasing the training data.

Third, weak physical interpretability. Deep learning-based methods are purely data-driven methods, which are limited in physical interpretability and overly dependent on the quality of the data used for training. Therefore, such problems can be solved by combining data-driven methods with physically driven methods.

With the accumulation of data and the development of deep learning techniques, the reliability of deep learning-based methods will gradually increase. Therefore, deep learning-based methods may become the mainstream of hyperspectral target detection algorithms in the future.

## 6. Conclusions

In this paper, we review comprehensively the target detection methods including classical algorithms and deep learning-based methods. After in-depth research, we divide the methods into seven categories and introduce the basic principles as well as classical and modified algorithms, respectively. We also give an outline of the datasets and evaluate metrics of target detection. We also analyze the relationship between the seven categories of methods and their advantages and limitations and experiment on the typical methods. Finally, we point out future challenges and directions. We hope this review will help related researchers comprehend target detection and start their research quickly.

**Author Contributions:** Conceptualization, Z.S., Z.Z. and B.C.; methodology, B.C.; validation, B.C. and L.L.; writing—original draft preparation, B.C. and L.L.; writing—review and editing, B.C., L.L., Z.Z. and Z.S. All authors have read and agreed to the published version of the manuscript.

**Funding:** This research was funded by the National Key Research and Development Program of China (Grant No. 2022ZD0160401), the National Natural Science Foundation of China under the Grants 62125102, the Beijing Natural Science Foundation under Grant JL23005, and the Fundamental Research Funds for the Central Universities.

**Data Availability Statement:** Not applicable.

**Acknowledgments:** We express our gratitude to all the editors and commenters for their valuable contributions and feedback.

**Conflicts of Interest:** The authors declare that they have no conflict of interest.

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
