# Peer review of "Target Detection in Hyperspectral Remote Sensing Image: Current Status and Challenges"

_remotesensing, doi:10.3390/rs15133223_

Round 1

Reviewer 1 Report

The authors conduct a comprehensive review of hyperspectral target detection algorithms. They summarize the existing hyperspectral target detection algorithms into seven categories, demonstrated the representative methods of each category, and analyze the differences and connections between methods in different categories. However, the manuscript still has some problems in content structure and writing. Detailed comments are as following:

1.      First, for the completeness of a review paper, the manuscript has some incompleteness:

(1) The authors classified the Deep learning-based methods into two categories, including "end-to-end detection" and "detection by reconstruction". However, we cannot find the classification basis and differences of the two categories. Please detail the advantages and disadvantages of the two types of methods.

(2) For the evaluation metrics, the authors introduced AUC of ROC and 3D-ROC, however, the calculation of AUC curve only stays in the literal description and is not clear. Please give the detailed calculation formula of ROC and AUC.

2.      Second, regarding the Future Challenge section, there are some problems:

(1) The relationship between 4.3.3 and other subsections in Future Challenge is confusing, and in fact, the problems pointed out in other subsections besides 4.3.3 are also encountered for deep learning-based methods.

(2) The Future Challenges section should give the authors' suggestions for future research directions. However, the authors have not suggest future research directions for subsection 4.3.4 "Causal real-time detection".

3.      Finally, there are some details about the writing:

(1) The second sentence of Section 2.1, "Let H0 represent the target is absent, and H1 represents the target is present", has grammatical errors.

(2) In section 2.1, there is no further explanation of v and a in Eq. (5).

(3) In order to ensure the correspondence of the paper, the method categories in Table 2 should be listed in the order of their appearance in the paper.

Minor editing of English language required.

Reviewer 2 Report

With the overall observation it is greatly appreciated for the amount of effort and analysis the author has put into this study. The manuscript is well written and presented. My major concern is the structure of the paper and comparative study, as I find this difficult to understand.

The recent year papers are less when compared to other papers. Include recent year papers (last 3 years).

Include the taxonomy diagram (like figure 2) in section 2. It will help the readers to understand the research effectively.

What is the impact or a purpose of Table 1, I recommend to include these dataset information in table 2 for a better understanding.  

The author used many variables, but the limit or the notation details are missing. Include it.

Fig 1,3 and 5: The visual quality is a bit low. Improve the content. Include cooke dataset image in the fig 1.

Reviewer 3 Report

The review, “Target Detection in Hyperspectral Remote Sensing Image: A Survey” (suggest deleting “Image: A Survey” from the title) summarized seven categories of hyperspectral image target detection methods. I really appreciated the thought and effort the authors put forward to summarize and compare the various categories in Figure 2, though I wished more detail describing how each algorithm was linked, as this figure was difficult to follow.

Overall, I thought the paper has potential, but it suffered on a few key fronts:

(1)    The structure of the paper should be changed. In review papers, it is not important to define the various functions/algorithms mathematically (section 2). You could however, include them in the supplemental materials, but the focus should be on maintaining readability and describing the important categories/algorithm’s function clearly and concisely. At present, the manuscript is difficult to follow and section 2 is not readable.

(2)    I was not sure what the aim of this review? Why do we need it? Why is it required? Just because other reviews (line 55) did not include ML and DL, does not mean we need another. Provide improved and clear justification for why such a review is timely and needed.   

(3)    This review paper does not adequately review the literature. In many places where a review of the literature should be (e.g. lines  241-262; 254-263; 320-347;386-395), instead a list of papers are provided, not comparing/contrasting, and the readers is left confused as to why this information is presented and what the authors are trying to argue/present. Good example of this is on lines 391-395, an array of new-shot learning methods were mentioned, without any context, the reader has no idea what is going on here other than many Siamese networks were created (what are they??).

(4)    Nearly all the text in the manuscript is difficult to read, lacks clarity, could be significantly simplified, is redundant in many places, and overall should be restructured as mentioned above.

I suggest: first provide clear and strong justification for this review. Determine the aim of this review (i.e. what do you want to accomplish). Restructure the entire paper around figure 2. Though there were sections briefly mentioning the various categories, much more detail is  needed describing what these are, how they are connected, and their strengths/weaknesses (in plain English, not mathematically).  Remove or move section 2 to supplemental. Remove simple lists of papers and actually review the literature with a set goal/aim in mind, which should come across to the reader. Best of luck!

NA
